# The Relevance of the Adverse Childhood Experience International Questionnaire to Working Children: Knowledge Gaps and Implications for Policy Makers

**DOI:** 10.3390/children8100897

**Published:** 2021-10-09

**Authors:** Alice Rutter

**Affiliations:** Hubert H. Humphrey School of Public Affairs, University of Minnesota, 301 19th Ave South, Minneapolis, MN 55455, USA; rutte079@umn.edu

**Keywords:** adverse childhood experiences (ACE), child trauma, child work, child labor, World Health Organization (WHO)

## Abstract

(1) Adverse childhood experiences (ACE) are a global challenge, prioritized in the United Nations’ Sustainable Development Goals. The ACE questionnaire is widely adopted in the USA as a tool for measuring population-level trends, such as negative health, behavioral, and economic outcomes. Intuitively, children in resource-scarce settings are exposed to higher levels of trauma. To understand the global picture, the World Health Organization (WHO) adapted the ACE international questionnaire (ACE-IQ), to inform policy and target interventions. However, evaluation of whether the ACE-IQ captures the experiences of around 160 million working children remains limited. (2) I applied the ACE-IQ scoring tools to detailed case studies of working children, comparing issues highlighted by holistic assessment to those captured by the ACE-IQ. (3) The ACE-IQ struggles to capture nuance across cultural contexts. As a consequence, application of the ACE-IQ as a policy tool risks “policy failure”. The tool reflects prevalent Western concerns, such as school attendance and parental supervision, but global concerns affecting working children such as forced economic migration and famine are neglected. This limitation produces “policy myopia”, sidelining certain global challenges. (4) The ACE-IQ is a useful public health tool, increasingly used to define policy goals. However, given the limitations of the ACE-IQ, the consequences of prioritizing these particular policy goals need to be actively acknowledged and mitigated.

## 1. Introduction

The global burden of childhood trauma remains tragically high, with the World Health Organization (WHO) estimating that 1 billion children between 2 and 17 years old are the victims of physical, sexual, or emotional violence per year [1]. The 2030 Sustainable Development Goal 16.2 targets ending violence against children, and a key part of this work centers on improving the global understanding of child trauma, and designing and implementing effective interventions [2]. One tool available to support this process is the Adverse Childhood Experience International Questionnaire (ACE-IQ), launched in 2011. Ten years after the ACE-IQ’s initial inception, this paper evaluates its usefulness as a tool for the world’s working children (around half of whom are described by the International Labour Organization (ILO) as highly vulnerable to harm) and examines the evolution of the ACE-IQ as a policy tool [3].

### 1.1. The ACE Questionnaire

The Adverse childhood experience (ACE) Questionnaire is a widely adopted tool for understanding childhood trauma, used in the United States (U.S.) for more than four decades. “Adverse childhood experiences” is often employed as a catchall term to describe childhood trauma, but the questionnaire aims to distil this down to discrete and quantifiable measures. The Centers for Disease Prevention and Control (CDC) reports extensive data demonstrating the correlation between ACE frequency, as measured by the questionnaire, and negative health and social outcomes [4].

A comprehensive review of the usefulness of the ACE Questionnaire was published in the Journal of the American Medical Association in January 2021. The researchers pooled data from Westernized nations including the United Kingdom, United States, New Zealand, and Norway to evaluate whether the ACE Questionnaire was a useful tool to guide healthcare intervention. While this study identified that at the population level, higher ACE groups were at higher risk of needing health interventions, the effect was so small that at the individual level the researchers concluded that the ACE Questionnaire did not offer greater insight than routinely available information—such as age and sex—and so concluded that it was not a helpful screening tool [5]. This limited individual benefit is important given that application of the questionnaire to individuals necessitates the emotional labor of revisiting traumatic experiences, and potentially harms those who feel that their traumatic experiences are invalidated due to exclusion from the ACE framework [6]. The value of the ACE Questionnaire is to demonstrate population-level trends; this suggests that the ACE Questionnaire exists as a way to understand trends and inform policy.

### 1.2. Application of the ACE Questionnaire as a Policy Tool

In 2019, the Centers for Disease Control and Prevention (CDC) announced a suite of high-level policy strategies to tackle the prevalence of adverse childhood experiences in the U.S., to complement their existing “technical” policy tools, which date from 2016 [5]. This reflects broader ambitions for more evidence-based global policy, and the ACE Questionnaire is a readily available tool to define the problem and determine tangible, quantifiable targets.

Reviewing the nature of publications in this field, Kelly-Irving et al. describe the growing emphasis on how the ACE Questionnaire can inform policy development for ACE prevention, perhaps in recognition of this opportunity for useful application. The authors note the increasing use of the ACE framework as a part of public campaigns and social movements [7]. However, ambiguity remains around the magnitude of the effect of each of the ACE domains.

Finkelhor, based in the Crimes against Children Research Center, discusses limitations of the ACE Questionnaire: what are we really asking when we use the ACE Questionnaire; what do we intend to do with the answers; and what are the possible negative consequences of asking [8]? Finkelhor concludes that there is great potential in using tools to identify children who need help, and strategically intervening to promote their future health. The hypothesis that there is potential for population benefit through improving the experiences of children offers promise, but Finkelhor caveats that as yet the evidence does not actually seem to support the use of the ACE framework for this purpose. There is a dearth of evidence that the questionnaire addresses the most pressing issues, that there is a causal link between the questions and adverse outcomes, that intervening to address these specific exposures will mitigate health risks, and that secondary risk reduction based on these measures is effective at improving long-term outcomes at the individual level.

### 1.3. Adaptation of the ACE Questionnaire for International Use

Following the publication of the WHO’s 2002 *World Report on Violence and Health* and the 2006 UN study on *Violence against Children*, there was increased attention on ACEs, and a desire to gather data to “inform policies and programs” [9]. For this reason, the WHO established an international ACE research network (IARN) to produce a “standardized international questionnaire” [9].

The IARN was established in 2009, led by the WHO’s Department of Violence and Injury Prevention and Disability; the WHO’s Department of Chronic Diseases and Health Promotion; and the U.S. CDC. The ACE-IQ is part of a wider health survey, but these are the measures to understand adverse experiences in childhood—building a picture of the impact of the measured exposures on health, behavioral, and social outcomes.

ACE questions have been expanded in ways reflective of member priorities, for example expanding the scope of domestic abuse to include either parent or guardian as a perpetrator and introducing a new domain around collective violence and displacement due to war. Other changes include a new question on bullying by peers, expanding the definition of unwanted sexual contact to include abuse by people of similar age, and broadening the neglect domain to include the extent to which parents have knowledge of, or understand, their child’s concerns and actions. Not sending a child to school is introduced as a form of physical neglect.

### 1.4. Synthesis of Existing Research into the Global Application of the ACE-IQ

In its 2011 report, the WHO published pilot studies testing the amended ACE International Questionnaire (ACE-IQ). These focused on how to adapt the wording of the questions to ensure that they were understood in different languages [9]. Initial field testing in China, Macedonia, Philippines, Thailand, Saudi Arabia, and South Africa assessed whether people understood the questions. However, there was no detailed exploration of whether the questions captured local experiences across these settings. A broader pilot study in Vietnam, surveying 2099 medical students, explored whether the raised mental health risk associated with higher ACE scores was replicated outside the U.S. context. This found that ACE measures correlate with worse mental health outcomes, and reported a dose–response relationship between ACE exposures and depression, suicidal ideation, drinking, and underage driving [9,10]. This study informs the IARN’s development of the ACE-IQ, serving as affirmation that the domains hold external validity across contexts.

A 2010 study in the Philippines found that, while over three quarters of adult respondents recalled ACE from their childhood (using an adaptation of the ACE Questionnaire), the majority of experiences fell into the categories of physical and emotional neglect [11]. This was surprising to the research team, as through their prior research, nearly half of this community had reported experiencing physical abuse as children—yet less than ten percent recorded this in the survey. Further enquiry suggested that the examples used in the questionnaire wording—such as “pushing” and “slapping”—did not reflect the experiences of local people [11]. Instead punishment by hitting children with a belt, or spanking children with hard objects was more common [11]. This dissonance between the wording of the question and interpretation through the lens of local experience led to massive underreporting of physical violence in childhood.

A cross-sectional study of 433 Chinese adults, using a translation of the ACE-IQ, found that ACEs correlated with negative health outcomes [12]. However, Ho et al. found that there was very limited internal validity in the domains of emotional neglect, emotional abuse, and aggression towards a member of the household [12]. Ho et al. concluded that the wording of the domains did not align with cultural norms and accepted practices, which meant re-interpreting childhood experiences through the lens of the questions resulted in variable responses—even by the same individual asked the same question at a different time [12].

To assess the need for cultural adaptation of the ACE-IQ, Quinn et al. tested the questionnaire with two focus groups in an underserved community in South Africa [13]. Undergoing a collaborative, iterative review process—with the community ultimately deciding what the questions should be—led to a significant increase in community participation with the ACE questionnaire survey [13]. However, cultural stigma around experiences of sexual violence meant this topic was particularly challenging to discuss in focus groups, with profound self-blame and fear [13].

A systematic review of the fourteen available studies of the ACE-IQ in low- and middle-income countries found that the proportion of children with high levels of exposure to adverse experiences was far greater in non-Western nations; 80% of individuals in Saudi Arabia had at least one ACE exposure [14]. Solberg et al. concluded that the correlation between ACEs and negative health outcomes means that using the ACE-IQ is helpful for public health surveillance [14]. Solberg extends this idea to argue that migrant populations in the US should be the focus of enhanced access to mental health and social services—which suggests a policy leap on the basis of ACE scores.

### 1.5. Working Children and the ACE-IQ

While the ACE-IQ includes additional questions on state violence and experience of war, other key global challenges are notably absent. Given that working children account for around one in five children, by conservative estimates based on economic productivity, working children cannot be subcategorized as a special case.

In 2020, the WHO highlighted that children’s existence is “at risk from rising sea levels, extreme weather events, water and food insecurity, heat stress, emerging infectious diseases, and large-scale population migration”, describing these as key threats to child health and security [15]. Given the magnitude of these challenges, and the extent of the effects reported by the WHO on children, it may be expected that these exposures are reflected as adverse childhood experiences. Rising economic migration and increased resource-scarcity, with accompanying inflation of costs of food and other goods, are likely to disproportionately affect the lives of working children.

An estimated 17 percent of children (under age 18) are engaged in labor-force work globally, according to the most recent data from the International Labour Organization (ILO), suggesting that for many children, work is an important aspect of the childhood experience [16]. It is important to note that international definitions do not recognize carer responsibilities or household work, resulting in an additional and largely hidden labor-force. Research in 2007 found that around two to four percent of children in Western nations were carers [17]. However, there is limited data available on child carers globally. This means that the actual number of children undertaking work is likely to significantly exceed the figure of 160 million reported by the ILO in 2021 [18].

Child work is highly relevant to the ACE-IQ as new questions integrated into the questionnaire, particularly regular school attendance throughout childhood and questions premised on the assumption that children are living in the family home are at odds with common childhood experiences in the global context. Child work may increase food security, reducing a child’s exposure to malnutrition and increasing their standing within the family unit [19]. As an example, a child may learn a trade instead of attending school, to earn so that they and their family can afford food and healthcare, thereby reducing their exposure to adverse experiences such as death of a close family member [19]. Eliminating exploitation and maltreatment of children, including working children, is crucial. But there is a need to be aware of counterfactual scenarios so that well-meaning interventions do not condemn children to a worse reality.

Of the 152 million working children (data from 2020), around 73 million are undertaking hazardous work [4]. Hazardous work is defined by the ILO as “work which, by its nature or the circumstances in which it is carried out, is likely to harm the health, safety, or morals of children” [4]. This suggests that many workplace experiences could be categorized as “adverse experiences” of childhood, and thus are important for the questionnaire to capture. As an example, it is common for young girls in Nigeria to be placed into domestic service, where they live in households in cities—geographically removed from their families who live in rural areas [20]. In many cases, young girls are exposed to physical violence at the hands of their employers [20]. However, maltreatment by employers or other workplace experiences are not considered as part of the ACE-IQ.

A search of Google Scholar, PUBMED, and the Web of Science did not find any existing studies examining the use of the ACE-IQ in working children; this is an important gap in the academic literature.

Despite existing reservations in the academic community, great interest in the use of ACEs has led to application of the ACE Questionnaire in contexts very different to the U.S. However, clear questions emerge from the existing literature and dialogue around the ACE-IQ. Firstly, is the ACE-IQ an effective tool to build a picture of adverse childhood experiences globally? Given the large population of working children and lack of clear consideration of their experiences either in the tool or the literature, I felt that there was value in considering the tool through the lens of this subpopulation. Secondly, and in the light of what the first analysis reveals, what does the use of the ACE-IQ mean as it becomes perceived as a policy tool? My analysis helps to bridge the gap in current understanding of how the ACE Questionnaire captures (or misses) the experience of children outside the Western context. Viewing the strengths and weaknesses of the ACE-IQ alongside the academic public policy literature offers insight into the policy implications of problem definitions and measurement tools, the potential impact of the global application of the ACE-IQ. This highlights areas where urgent work is needed.

## 2. Materials and Methods

I took a multi-phased approach to fully appraise the usefulness of the ACE International Questionnaire (ACE-IQ).

### 2.1. Working Children Case Studies

To assess the usefulness of the ACE-IQ in describing traumatic experiences of childhood globally, I applied the ACE-IQ binary scoring framework (and using the scoring guide published by WHO online) to detailed case studies. The full scoring tool was used including all questions, and the high-level domains were summarized for presentation purposes.

These case studies were selected with the support of expert academics in the field of child labor, based at the University of Minnesota. Each case study was selected on the basis of:(a)Offering a distinct cultural or workplace perspective;(b)Containing sufficient detail to allow a holistic approximation of the ACE-IQ score; and(c)Specific recognition by the author of cultural nuance, especially where the authors were not writing about their own cultural experience.

After a broad review process, review of cases with peers, and consultation with in-field experts, I selected three detailed case studies from the academic literature—each looking at experiences of working children in different cultures, contexts, and stages of childhood. Each of these case studies was reviewed in detail, using a two-phased approach. The first review drew out a holistic impression of the factors that were relevant to the cases considered, with reviewers’ notes taken contemporaneously and reflections discussed at the time with senior academics in the field. A second, detailed review was undertaken to document and record all text identified as relevant to the ACE-IQ binary scoring guideline. This was used to approximate a summary score, and elements taken from the text are used to illustrate the scoring rationale. My holistic impression and the technical scores are both presented, and areas of contrast highlighted.

By taking this approach, I was able to benefit from authors who had gone into great depth in understanding and describing child experiences qualitatively—in sufficient detail to approximate ACE-IQ scores. This highlighted both areas that were captured, as well as key experiences in that context that were significant but absent from the ACE-IQ.

### 2.2. Critical Review through the Academic Policy Lens

I brought together articles, meeting reports, and policy documents published by the WHO to build a picture of the approach to the development of the ACE-IQ, types of evidence used by the team convened by the WHO, and the policy goals for this piece of work. Drawing from my own academic study in the field of policymaking, policy analysis, the politics of policy processes, and collaborative approaches to effective health policy, I brought together relevant literature in the field as a framework to analyze the ACE-IQ. Building upon the findings of the first part of my analysis, the case studies, I employ the public policy literature to offer critical insight into what these findings mean—and what the field of public policy science would suggest as possible remedies for any shortcomings identified.

## 3. Results

### 3.1. Case Studies from the Literature

Drawing case studies from the academic literature offers insight into how the ACE-IQ distils the diverse experiences of working children down to quantifiable measures. The case studies selected, presented by academic writers seeking to capture childhood experiences, offer detail into many aspects of children’s lives beyond the scope of the questionnaire. These authors have undertaken extensive—and, in some cases, immersive—research to gain a full understanding of cultural and social complexities, which is helpful in gaining insight into contextual priorities and how these align with the ACE-IQ. Using case studies from existing literature offers access to detailed and diverse accounts, but also meant that experiences were captured by a person who had already gained the trust of these children. I felt that this was crucial, given the sensitive nature of the questions within the ACE-IQ, to building a clear picture of the ability of the questionnaire to quantify childhood trauma as different children and communities perceive it.

The three case studies were not selected because of exposure to specific traumas, but rather as detailed and complex portraits of global childhood experiences—written by authors immersed in the social and cultural context. The case studies offer insight into experiences of children working in agriculture, industry, and the service sector. Agriculture is by far the most common type of child work globally; the ILO reports that agriculture accounts for about 71 percent of the 152 million children working globally [16]. About 12 percent are in industry, and 17 percent in the service sector [16]. The detail in these studies is used to make an approximation of ACE scores. Using secondhand accounts, I cannot make assertions about the lives of individuals or how they would answer the questionnaire. However, by taking this approach, I hope to offer insight both into the relevance of the ACE-IQ questions across cultural contexts and offer the first critical assessment of whether the ACE-IQ reflects the experiences of working children.

### 3.2. Children in the Chillihuani Region of Peru

Growing Up in a Culture of Respect by Inge Bolin (2006)

Bolin presents an account of children growing up in a remote village in Peru, in a close-knit indigenous community that depends on agriculture and subsistence living. The community has a deep spiritual connection to their land and animals, and children are expected to contribute to the communities’ way of life.

Applying the ACE-IQ to Bolin’s account of childhood experiences in Peru highlights the traumatic impact of the loss of a guardian (due to high mortality rates), and discriminatory experiences for children that leave the village. A high proportion of children do not attend school despite it being available, in part as it is four hours’ dangerous walk away. However, children are offered apprenticeship-style training within community roles. These challenges would increase the ACE-IQ score of the Chillihuani children (Table 1). Additional key stressors in this community are not captured. The land and its creatures, though holding enormous cultural importance to many indigenous communities and being fundamental to the Chillihuani belief systems, are not recognized as a potential source of traumatic events in the ACE-IQ.

The Score in Context

Culture: Bolin describes how, on first encountering the children living in the high-altitude Chillihuani village in Peru, she struggles to see “how survival could be possible” (p. 1) given the exposure to such an extreme environment and only the most basic tools to aid in subsistence from the land [21]. However, Bolin comes to appreciate the “care, respect, and compassion” (p. 1) that defines the community’s ideology—extending not only to all people within the village, but equally to the land, animals, and objects that support their survival in such marginal conditions [21].

Education: Around half of the village children can participate in formal schooling, but their society offers traditional learning through observation and increasing levels of responsibility and trust. Those that attend school in the valley, and that go on to universities, are “always at the top of the class” (p. 155) with a particular talent for mathematics [20]. Children start school around seven to eight years old, walking up to four hours across challenging terrain and in difficult weather conditions to attend (p. 85) [21]. Bolin notes that, for indigenous children, schooling can often serve a traumatizing “civilizing” (p. 86) purpose [20].

Work: Bolin describes the contrast between the Chillihuani vision of paradise, a place of agricultural plenty where there is work for all—including children—and the Western vision of paradise as a place of eternal leisure. Local children see paradise as “a place where hard work brings good results” (p. 72) [20]. Children leave to work in the high pastures in all conditions, including thunderstorms, hail, and snow (p. 76) [21]. Children’s activity is a major contribution to their community’s subsistence, it is valued work that brings them closer to the deities, which the children take pride in doing, and is considered “fun” (p. 157) [21]. From fourteen, children start apprenticeship-style training for roles within the structure of their community (p. 145) [21].

Family: Children are appreciated by their families for the “help and support they provide” (p. 57) as part of the subsistence lifestyle, but even when children leave the village they are “loved and always welcomed” (p. 57) [21]. Children are “the center of attention” (p. 56), and “never neglected” (p. 56) [21]. Disabled children are carefully cared for; the story of one deaf child tells how he experienced discrimination for the first time when he left the village to work in the city—both as he was from the mountain community, and due to his disability.

Stress: Sources of stress in the community include bad harvests, death or sickness in the family, extreme poverty, and threats from extreme weather and wild animals (p. 141) [21]. Difficult economic conditions increasingly mean that both adolescents and adults leave the community to look for work, including children as young as fifteen. Experiences of discrimination are common and traumatic outside of the confines of the Chillihuani village, and exposure to new diseases and malnutrition means that many villagers die (p. 141) [21].

### 3.3. Children in Post-War Afghanistan

The Bookseller of Kabul by Asne Seierstad (2002)

In *The Bookseller of Kabul*, Seierstad presents her account of living with a family in post-conflict Afghanistan. The story presents Seierstad’s observations of the family dynamics, as well as reports of discussions with various family members. Mansur and Leila’s experiences as children growing up in postconflict Afghanistan are detailed throughout the book, and by drawing together these aspects of their stories I have developed a fuller picture and estimated an ACE-IQ score (see Table 2).

Both Mansur and Leila are exposed to a range of adverse experiences, including destruction of their home and the violence of a protracted war. They are forced to flee their country as refugees. However, many of their experiences are centered on a strict and hierarchical family structure, which means that they feel significant personal insecurity—as their status and acceptance within the family are frequently threatened. They come to resent the limitations on their choices and available opportunities. This absence of personal empowerment is emphasized as the source of great sadness and turmoil.

The Score in Context

Culture: Much of the description of the lives of Masur and Leila focuses on their return to Afghanistan after the removal of the Taliban regime. While the family was “middle class” (p. 15) with “enough money” (p. 15) and “never hungry” (p. 15), “half of Kabul had been reduced to a pile of rubble” (p. 18) and the evidence of destruction is everywhere [22]. Society is painted as deeply religious, strictly patriarchal, and with an emphasis on rules. The father asks, “if families don’t have rules, how can we form a society that respects rules and laws, and not just guns and rockets?” (p. 286); “scoundrels cannot be let loose” (p. 286), and punishments are firm [22]. There is a description of how a girl’s mother “dispatched her three sons to kill [their sister]” (p. 36) after she was seen with a man that was not her husband [22].

Education: Under the Taliban, education of women was prohibited and Leila continues to self-impose this ban after the change of leadership, feeling “dirty, exposed, her honor impaired” (p. 183) in a school with boys [22]. However, Leila’s education as a refugee in Pakistan means that her English is good enough to qualify as an English teacher. Her family’s decision is that she will marry, and it will then be at the discretion of her husband as to whether she can teach. Mansur “finished only ten classes” (p. 134) when his father took him out of school, prioritizing the development of the family business over his son’s education [22].

Work: Mansur feels that his father “chains him to the shop” (p. 131), and treats him as “a slave” (p. 132) [22]. Mansur watches his father’s bookstore, with responsibilities including cataloguing books, managing stock, and supervising other workers. When Leila nears the age of 19, she attempts to find work as a teacher, which is something she sees as a path to freedom and independence. However, she grows frustrated at how the bureaucracy prevents her finding work. “But you don’t have an English teacher ... can I start now and apply later?” (p. 191) she asks [22]. “Impossible. You must get personal clearance from the authorities” (p. 191) [22]. She knows that she will not be able to do this without the patriarch discovering; fearing “he would put his foot down” (p. 192), she has “reached a deadlock” (p. 193) [22].

Family: Seierstad paints the image of a strict patriarchal society, where the senior male member of the family’s “word is law” (p. 114), and challenging this absolute authority “will be punished” (p. 114) [22]. Derogatory comments about women, such as “parasite” (pp. 167, 179), “peasant girl” (p. 65), and “stupid as an ass” (p. 65), are commonplace [21]. When Mansur’s father reaches middle age he decides to take a new, teenage bride—which his mother finds a “shaming” (p. 10) experience, and for many years she forcibly remains in Pakistan while the rest of the family return to Kabul [22]. When Mansur’s cousin, Fazil, is dismissed from the family home without explanation–“I’m fed up with you. Go home” (p. 187)—the rest of the family are affected by a sense of insecurity [22].

Stress: Both Leila and Mansur feel constrained by the expectations that their family place upon them. Leila dreams of having a job and meets a potential husband that will allow her to achieve that goal. When the family decides that she will marry within the family instead, and “remain the servant girl” (p. 274), Leila “feels her heart, heavy and lonely like a stone, condemned to be crushed forever” (p. 282) [22]. As a younger man, Mansur works to please his father. As a teenager, he comes to regret his choice to report a theft to his father when he tries to reconcile with the consequences of his actions. “[H]e might get six years! His children might be dead when he gets out” (p. 237), he shouts at his father [22]. Still, he is obliged to take the man, accused of stealing postcards, to the jail (p. 237).

### 3.4. Female Garment Factory Workers in Bangladesh

Transition to Adulthood of Female Garment-Factory Workers in Bangladesh by Amin et al., Studies in Family Planning, Jun., 1998, Vol. 29, No. 2, Adolescent Reproductive Behavior in the Developing World (Jun., 1998), pp. 185–200

In this account, Amin offers qualitative insight through interviews with young women working in garment factories in Bangladesh. The experiences are diverse, but Amin highlights key themes that recur, and common shared experiences and attitudes. Using Amin’s analysis, and drawing from detailed examples and quotes, I have estimated the ACE-IQ score as around 2 (see Table 3).

Unaccompanied travel and living away from the family are a source of stress for the young women. However, while the young women are exposed to potentially exploitative working conditions, this exploitation is seen as a route to greater economic and personal freedom compared to their peers [23]. Risks in one domain, for example keeping girls in school and away from negative work environments, increase the likelihood of early marriage and early motherhood—carrying a high risk of mortality.

The Score in Context

Culture: Amin et al. describe how in Bangladeshi culture, as girls enter their adolescence and develop the physical characteristics of womanhood, emerging sexuality is conventionally managed by marrying off young women and undertaking “purdah” (p. 186)–whereby interactions between men and women are tightly regulated [23].

Education: Around half of workers in the Bangladeshi garment factories have no formal education (p. 191). Women working in the factories are gaining skills in work but are not in specific training roles (p. 191).

Work: The garment factories in Bangladesh are the first places to offer employment to young, single women from rural areas (p. 185). Wages for women are lower, even though in the factories they are doing the higher-skill work (p. 186). Amin emphasizes that “exploitation and liberation go hand in hand” (p. 187) in this setting [23]. These women face stigmatization by their peers and are characterized as sexually promiscuous because they work (p. 188). The young women typically start work between thirteen and sixteen years old, leaving school either for financial reasons or due to low school attainment (p. 189). The women report that they either made the choice to start working or were involved in the choice (p. 190). Work in the garment factory involves either manual work, such as cutting fabric, or higher paid work as machine operators. Women move between factories to increase their salaries (p. 193).

Family: For many of the garment workers, working in the cities means living separately from their families (p. 194). In large part, the women can retain their income, which offers a degree of independence (p. 188). In some cases, women save some of their income to increase their dowry, offering them more choice of husbands as well as retaining a higher degree of independence in married life (p. 194). Through work, women build strong social connections with their co-workers (p. 185).

Stress: The garment factory workers face stigma within their communities, and experience fear travelling to and from work (p. 190). Working in the factories exposes the women to harsh working conditions and long hours, which manifests as ill-health such as worsening eyesight, diseases, and significant weight loss (p. 195). However, it also reduces some pressures that young Bangladeshi women face, by enabling the workers more agency in their marriage and delaying childbearing (p. 185). This is particularly important given the high risks associated with early motherhood (p. 199).

## 4. Viewing These Findings through the Lens of the Academic Policy Literature

The policy literature offers insight into potential secondary effects arising from limited scope, and challenges of cross-cultural relevance, encountered by the ACE-IQ.

### 4.1. The Policy Aims behind Extending the Use of ACE-IQ

The WHO International ACE Research Network’s (IARN’s) purpose is to reduce adverse experiences for children globally. However, IARN’s focus quickly turns to measuring adversity, and applying this to evaluate the effectiveness of policy interventions. The IARN sees the ACE Questionnaire as the practical way to achieve this end—as it is a tool that exists, with extensive prior research in the U.S. The WHO states that “[the] standardized ACE-IQ will enable the measurement of childhood adversities in all countries and comparisons of such adversities between them; the drawing of associations between childhood adversities and health risk behaviors and health outcomes in later life; advocacy for increased investments to reduce childhood adversities, and scientific information to inform the design of prevention programs” [9]. Additionally, it is likely to be in the interest of the U.S. stakeholders within the IARN to extend the use of a familiar tool, that they already use routinely, and which has been the focus of research supported by U.S. institutions such as the CDC.

The IARN does not set out specific policy objectives. However, in establishing the parameters that measure success and access to funding, there is a risk of creating a rigid framework for policy design. The stated purpose of the IARN does recognize this consequence, noting that the measurements will “inform the design” of programs.

### 4.2. Stakeholder Inclusion in Policy Design

The policy literature considers including the relevant perspectives as part of a design process, such as the ACE-IQ design, crucial to ensuring that the right issues are prioritized and that there is effective buy-in to solutions. Bryson et al. describe how in order to engage in effective discussions about producing high-quality plans, policies, or programs, there needs to be a deliberative process that includes a range of voices, including experts [24]. There are multiple ways of understanding a problem; it is important for policymakers to recognize that diverse types of knowledge are useful [24].

Collectively defining the aim, indicators, and outcome measures, is an important first step in a collaborative process [25]. Agreeing the intended group purpose sits at the core of subsequent work in successful collaborative networks, holding group members accountable to the shared aim, and enabling effective collaborative governance [25]. The IARN is clearly a collaborative structure. However, the scope of IARN membership as reported by the WHO in 2011 is limited. While this may be practical, to enable rapid decision-making and streamlined processes, Bryson et al. suggest it is likely to reduce stakeholder buy-in as groups are brought in late in the process [25].

### 4.3. Risk of Policy Failure

Public policy literature highlights the common features of policy failures. McConnell describes how outcomes failing to address the needs of the intended beneficiaries, or not meeting the intended policy aim, are likely antecedents of policy failure [26]. Bardach describes the eightfold path to effective policy design, with step one being to carefully define the policy problem to be solved, with inadequate attention to problem definition causing “policy myopia” [27,28]. Policy myopia is a narrow framing of the problem such that policy outcomes cause active harm to the target population, by failing to appreciate the full extent of the issue [27]. In particular, Nair and Howlett describe how not acknowledging that the scope of a policy problem has been artificially limited can lead policies to fail due to poor implementation, uncertainty, and failing to understand the effects of interventions [27].

### 4.4. Embedding Targets

Moynihan and Soss discuss how once policies are introduced they become self-perpetuating, as policy feedback further entrenches the original political intentions and definitions [29]. The policies themselves, even those more technically focused such as measuring childhood adversity in this case, are central to defining the political processes and goals that follow. The original policy determines “who gets what, when, how”, and conveys to the public how they should understand the issue [29].

As the performance of bureaucrats on the ground, or even organizations and governments more broadly, becomes defined by the metrics outlined in the original technical policy outline, the original measures are embedded as targets to be met rather than one aspect of a more complex picture [29]. The more that individuals frame their success within the context of these outcomes, the greater the incentive to ensure that these outcomes continue to be valued and prioritized. In fact organizations and bureaucracies are increasingly motivated to protect the measures that they are using and therefore vigorously defend their importance [29].

## 5. Discussion

### 5.1. Limitations

The analysis of the ACE framework and how it has been adapted is based on available published materials; it is possible that the IARN has taken a different approach, beyond the scope of information I have available. The most up-to-date version of the ACE-IQ is not the version in the 2011 WHO Report—the question around school attendance was subsequently added. Additionally, I have not touched upon the organizational politics of the CDC, the WHO, the UN, and other institutions within this policy process. These politics may well form a significant element of how the outcomes are framed.

In considering these case studies, I have selected three examples from the literature after a detailed review process to reflect different cultural settings and types of child work. Given that these examples are based on in-depth studies by researchers in the field, I have assumed that these are reflective of a certain setting at a certain moment in time. However, I cannot make assertions about the actual lives of the children involved and would not wish to do so. I have no way of knowing how representative the experiences of the children described are of children within their wider society or the world more broadly. In addition, the literature surrounding the global application of the ACE framework suggests that the wording of the questions is interpreted differently across contexts, especially around culturally sensitive issues, and I am unable to reflect that nuance here. The intention is purely to employ these detailed accounts to consider how well the ACE-IQ framework can speak to the traumatic or challenging life experiences that these children have reportedly encountered.

These case studies are were published in 2006, 2003 and 1998 respectively. The situations described, the challenges of preserving indigenous culture, work in garment factories away from home, and experiences of war in Afghanistan still feel relevant to me in this moment—perhaps even more so with the capture of Kabul by the Taliban in 2021. Additionally, as the ACE-IQ is often used to capture retrospective rather than contemporaneous experiences—even if the policy landscape has evolved so extensively that the challenges faced by the children in these case studies are no longer current, the policy implications still stand.

My own frame of reference, growing up in the UK and studying in the U.S., is in-keeping with a relatively typical Western childhood. I have sought to actively challenge that this perspective is the only way to understand child trauma and ACEs but appreciate that my understanding of the case studies is likely to be impacted by my own cultural frame of reference.

### 5.2. Strengths and Weaknesses of the ACE-IQ as a Tool for Working Children

The ACE-IQ captures experiences of abuse due to actions or inactions of parents and guardians, such as the verbal abuse Mansur and Leila experience. However, it is not designed to capture experiences such as those of the Bangladeshi women—who live primarily away from the family home for much of their childhood. Domestic work outside of the family home is still a common experience for children, especially girls, globally. Confining both abuse and neglect to parents and guardians means that the experience of children at work, and the experiences of children who live away from the family home, are not fully captured by the ACE-IQ. The number of children who are undertaking “hazardous” work is estimated to be around 73 million. The ILO defines hazardous work as harmful physically, psychologically, or morally. As a consequence of limiting the scope of abuse to household members, it is likely that using the ACE-IQ would result in a significant underestimation of global childhood trauma, and a failure of the ACE-IQ to recognize a large population at potential risk of the secondary harms associated with ACEs.

Parents not sending their children to school when it is available is classified within the ACE-IQ as a form of physical neglect. This is a controversial inclusion, given the criticisms within the literature that preference of schooling above experiential learning, apprenticeships, and developing through work is highly reflective of a Western ideology, seeing childhood purely as a time of innocence and dependence [19,30]. Even in the U.S., apprenticeships are increasing in popularity as an alternative to conventional schooling [31]. For young women in Bangladeshi factories, leaving school to find paid employment increases economic independence and reduces the risks associated with early marriage and pregnancy. The children in the Chillihuani village undertake continuous training and apprenticeships to prepare them for life within the community, enabling them to contribute to village life. This experience is valued by both the children and community and presents an alternative to long and dangerous journeys to the local school. Defining this as neglect seems out of step with how the community perceives it. But beyond this, the classification of poor school attendance as neglect sits in tension with the possibility of exposure to greater harms in counterfactual scenarios confronting these children.

The inclusion of bullying in the ACE-IQ is a step to better understanding the role of peers in childhood trauma. However, it is unclear whether experiences of racial discrimination or religious persecution, such as the mocking of children’s beliefs experienced by the indigenous people in Chillihuani, would be classified as bullying. Infringement of fundamental human rights through racial or religious discrimination does not feel appropriately encompassed by the term bullying, and defining bullying as verbal action by young people suggests that this is not the intended meaning. International organizations such as the Cato Institute report that religious persecution remains a global problem, with 56 nations imposing very high restrictions on freedoms and rising numbers of attacks motivated by religious beliefs [32]. The United Nations (UN) reports that racial and ethnically motivated persecution is a daily occurrence, hindering the progress of millions of people [33]. The lifelong impacts of cultural persecution in childhood are increasingly recognized; the Truth and Reconciliation Commission into the Indian Boarding Schools in North America from 1860 to 1978 resulted in the recognition of deep multi-generational trauma and need to compensate affected Native American people [34]. 

Introduction of questions around community and collective violence highlighted Mansur and Leila’s experiences of the war in Afghanistan. The focus within these domains is on directly experiencing acts of physical violence by people in formal positions of power. Experiencing violence at the hands of an employer would not generate a score given that the question specifies “soldiers, police, militia, or gangs”. While forced escape from conflict is within the scoring system, highlighting the experiences of Mansur and Leila in Afghanistan, other reasons for migration would not be defined as ACEs. As is seen in the Chillihuani village case study, climate change and famine can force economic migration—and the impact to children of being forced from the security of their own communities can be profound. Exposure to new diseases and harsh conditions, without the security of a supportive community, means that Chillihuani people die seeking economic opportunities. Likewise, some young women in Bangladesh leave their families to seek economic opportunities. Forced migration, irrespective of cause, seems to have a significant impact on the lives of many children. For the Chillihuani people, the ideological significance of their environment to their belief systems further compounds the trauma of having to leave for survival.

The ACE-IQ does identify many experiences of working children that have potential to be traumatic. However, it has significant blind spots, in particular when considering specific experiences related to employment, economic migration, or living away from the family home. The initial studies considering the application of the ACE-IQ in diverse cultural settings identified challenges relating to the content, format, and nature of the questions. It seems that this is also the case for working children, whose experiences are only partially captured.

### 5.3. What Does This Mean for Policymakers?

It is possible that in focusing on technical tool design, the international ACE research network (IARN) does not see itself as setting the policy agenda. However, the wording of the report from the WHO published in 2011 suggests the impact of the tool upon policy, and that its function within policy and program design, is recognized.

The public policy literature offers insight into the ideal approach to designing policy tools. Bardach et al. set out the eightfold path approach, which is premised upon a careful consideration and shared definition of the policy question before embarking on a policy process [28]. Bryson et al. describes the importance of stakeholder inclusion in this process of policy design, with a shared definition of the problem being the foundation of cross-sectoral buy-in to solutions [25]. Given the broad reach of the domains within the ACE-IQ, from education to global conflict, the field of stakeholders is potentially unwieldy. Trying to assimilate such a diverse range of perspectives could lead to stagnation due to competing interests, especially without clear stewardship. With the financial and policy endorsement of large and well-respected international actors such as the WHO, the IARN is actually well-positioned to collaboratively design policy tools to increase global understanding of child trauma and sustain their momentum. However, the IARN is largely not in a position to exert influence on global actors who do not agree with their understanding of childhood adversity. Policy is an inherently political process, and fostering a more collaborative approach may yield greater returns for the IARN as they seek to translate acquired knowledge through use of the ACE-IQ to positive outcomes for children.

The policy literature highlights how policy myopia can result in policy failure. Nair and Howlett describe how failing to pay adequate attention to policy definitions can lead to policy that is difficult to implement, and aims can get lost [27]. By using an existing tool to inform the scope of policy goals, rather than defining the goals prospectively, there is definitely a risk that the scope of the problem has been too narrowly framed. However, the application of the ACE-IQ framework to case studies of working children showed that experiences considered as traumatic were captured in many instances. Designing interventions that target the domains within the ACE-IQ would tackle the negative experiences of working children, and so this is not an inevitable policy failure. Indeed, in some ways, the fact that many of the experiences of working children were captured by the ACE-IQ tool suggests successful design.

Still, the limited scope of inclusion in designing the ACE-IQ does pose a significant challenge to its application as a policy tool. Soss and Moynihan describe the process of internalization of policy goals into multilevel bureaucracies, and how these rapidly become established as targets—often to the detriment of other priorities [29]. What this means for children is that once policy definitions and measures are embedded, through the ACE-IQ, these parameters will define perceptions and action to tackle child trauma for many years to come. This cycle is an incredibly difficult one to break. That it is being endorsed by the largest global health organization in the world, as well as key funders, makes it particularly difficult for alternatives to emerge.

## 6. Conclusions

There is a tension that policymakers and the field of public health frequently reckon with, between employing a tool that is known to be imperfect but which is readily implementable, and committing finite resource to a process of designing and implementing something new (and thus delaying implementation). Of course, there is never an absolute guarantee that the new tool will prove more effective than its predecessor. Progressive globalization and increasing international co-ordination of both policy interventions and measures makes finding common ground pragmatically necessary. The 2030 Sustainable Development Goal to “end abuse, exploitation, trafficking and all forms of violence against and torture of children” adds a sense of urgency to both understanding the scope of the problem, and finding effective policy solutions while there is increased international focus.

It is not my intention to assert that the ACE-IQ cannot offer useful insight into global experiences of childhood trauma and adversity. Many of the measures within the ACE-IQ address urgent global challenges, such as the exposure of children to police violence and war. It is not feasible to build a public health screening tool that holistically captures the experiences of each individual, and to some extent, compromise on content is inevitable. However, in the case of the ACE-IQ the policy problem has been viewed through the prism of priorities already embedded in the ACE framework, allowing these to suffice rather than critically re-examining the questionnaire’s foundations. I would challenge whether there is meaningful recognition of the significance of embedding these measures, and secondary effects of doing so.

The policy literature emphasizes the defining role that policy measures play in determining future policy directions and outcomes. The limited field testing reported by the IARN, beyond checking that the questions can be understood; narrow scope of consultation; and the absence of consideration of large sub-populations such as working children suggest that there has not been due attention to making sure that these measures and definitions are the right ones. This matters, because in not optimizing the utility of the measures as far as practical within the scope of resource constraints, the effectiveness of interventions is unnecessarily compromised. Policy and program designers are more likely to design interventions that align with these established measurement tools, and to dedicate time to making demonstrable progress against these measures; and funders are more likely to look to projects able to demonstrate success against these parameters. The policy literature warns that embedding targets in this way, regardless of good intentions of continuing to take the wider context into account, defines what comes to be practically important. In short, getting the policy measures wrong sets the international community up for policy failure. There is a risk that in focusing the world’s attention on a narrow range of issues, other, more urgent, challenges where intervention may be more impactful are neglected.

In practical terms at this stage, replacing the ACE-IQ may not be the best alternative. Doing so could be criticized as a reactionary disregard for something of value in preference of something that does not exist. However, further research and broader consultation to understand the limitations of the ACE-IQ and how to effectively integrate these parameters into policy measures is clearly needed.

## Figures and Tables

**Table 1 children-08-00897-t001:** ACE score for children in the Chillihuani village.

ACE Category	Description of Experiences	Relevant ACE-IQ Questions	Summary WHO Binary Score (Out of 13)
Abuse	Adults are expected to model positive behavior for children; aggression or violence is exceptionally rare.		No score.
Household challenges	Death of family members due to exposure to malnutrition or extreme cold is not an uncommon experience; economic migration exposes children and their families to new diseases that can be fatal.	Did your mother, father, or guardian die?	YES = 1
Neglect	Children are treated as adults and included as full and productive members of the community. However, around half of children do not attend formal school.	Did your parents/guardians not send you to school many times even when it was available?	YES = 1
Bullying	Respect is emphasized as a way of life, becoming the “very nature of a child” (p. 160). Bullying is not tolerated in the Chillihuani culture; however, the villagers who leave describe discrimination due to the perception that they are “simple” mountain people (p. 141).	Were you bullied many times?	May score if leaves the community, but whether the experience of discrimination would be interpreted as bullying is ambiguous.
Collective or community violence	Violence within the community is exceptionally rare. Death and destruction, or economic migration, caused by natural disasters or animals does not score.	Did you hear or see someone being beaten up in real life many times?	May score if leaves community.
Total			Likely range of scores 0–4

**Table 2 children-08-00897-t002:** The Bookseller of Kabul.

ACE Category	Description of Experiences	Relevant ACE-IQ Questions	Summary WHO Binary Score (Out of 13)
Abuse	There is no description of physical abuse of either Mansur or Leila. However, insults and humiliation as well as threats should they not follow instructions are common. Mansur fears he will be “disinherited, thrown from the house” (p. 240) if he defies his father. There is no sexual abuse of Leila or Mansur, however, Mansur struggles with being a bystander to the rape of a young girl (pp. 127–128)	Did a parent, guardian or other household member yell, scream or swear at you, insult or humiliate you?	YES = 1
Household challenges	There is clear description of insults directed at women within the household, and description of how Mansur’s mother felt shamed by the arrival of a much younger new wife in the house (p. 8).	Did you see or hear a parent or household member in your home being yelled at, screamed at, sworn at, insulted or humiliated? Were your parents ever divorced or separated?	YES = 1 Additional 1 if counting physical separation of parents.
Neglect	Both Leila and Mansur struggle to reconcile themselves with the path determined for them by the family patriarch. When Leila is told of the marriage planned for her, she “feels how life, her youth, hope leave her–she is unable to save herself” (p. 282).	Did your parents not send you to school many times even when it was available? Did your parents rarely or never understand your problems and worries?	YES = 1
Bullying	Bullying is defined as by other young people, and so is not described directed at Mansur or Leila.		No
Collective or community violence	Mansur’s home was “pillaged and burned” (p. 111) during the conflict that escalated in 2001. Mansur and Leila both flee Afghanistan with their families to Pakistan for much of the conflict. The descriptions of violence make it very likely that Leila and Mansur would have witnessed violence in their community many times.	Were you forced to go and live in another place due to any of these events? Did you experience the deliberate destruction of your home due to any of these events? Did you see or hear someone being beaten up in real life many times?	YES = 2 (for community and for collective violence)
Total			Likely score 5–6 or higher

**Table 3 children-08-00897-t003:** Garment factory in Bangladesh.

ACE Category	Description of Experiences	Relevant ACE-IQ Questions	Summary WHO Binary Score (Out of 13)
Abuse	There is no description of abuse by parents, especially as most of the young women reported that they were involved in the choice to leave home to find work. However, there is a suggestion of abusive work practices and economic exploitation.		No
Household challenges	There is no description of this; however, the girls frequently left the family home to work in urban areas.		No
Neglect	Given that the girls were living away from their parents, it seems likely that their parents may not have known what they were doing in their free time. Additionally, parents did not send their children to school when it was available.	Did your parents/guardians not send you to school even when it was available?	YES = 1
Bullying	These girls faced peer discrimination, being characterized as sexually promiscuous.	Were you bullied?	YES = 1
Collective or community violence	Girls moved away from home due to economic necessity, in the hope of improving their lives. This is not reflected in the scoring.		No
Total			Likely score around 2

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
