# Peer review of "The Relevance of the Adverse Childhood Experience International Questionnaire to Working Children: Knowledge Gaps and Implications for Policy Makers"

_children, 2021, doi:10.3390/children8100897_

Round 1
Reviewer 1 Report
This intriguing manuscript challenges the application of WHO’s adapted Adverse Childhood Experiences International Questionnaire to global populations, identifying limitations in the research base of the original ACE measure and questioning the generalizability of selected adverse experiences. This is an important topic of broad interest and in need of rigorous research. Unfortunately, I did not feel this manuscript was sufficiently systematic or scientific in its approach. I found the organization quite difficult to follow, as literature review and authorial arguments appeared throughout the results section. The methods were not described in detail and I did not feel this use of “case study” was appropriate or adequately cited. As a reader, this manuscript seemed more to me like a persuasive essay written in support of a strong thesis than a scientific article. I am including more specific comments below that may help to inform another version.
Abstract:
- The final two sentences in the abstract (labeled number 4) do not seem to fit together.
Introduction:
- Page 1 - I disagree that the ACE questionnaire is a “pre-eminent” tool – I think it is very frequently used for its brevity and perceived ease of administration, but don’t think it is necessarily seen as the most rigorous option.
- Page 1 - I recommend avoiding causal language regarding the associations of ACEs with later health outcomes, as these are correlational findings and may reflect third variables.
- Page 2 – Please provide evidence supporting the argument that responding to the ACE questionnaire is “potentially traumatic.”
- Pages 2-3 – Much of this section articulates a clear authorial position without adequate empirical support or context (e.g., asserting without citation that centering the ACE questionnaire leads to neglect of other domains, referencing an argument rather than evidence that the ACE measure has been insufficiently adapted, etc.)
- Page 3 – The authors make many good points about the role of work cross-culturally, but their main point is not clear. Are they arguing that work may be an adverse experience, protects from adverse experiences, or both? What is the implication for their arguments about revising the ACE-IQ? It is not clear that the authors are trying to prepare the reader for a focus on children who work.
Materials and Methods:
- Page 3 - Methods were not at all clear to me. How were articles identified and selected? If this was a systematic review, authors should adhere to PRISMA reporting requirements. It was not clear how “case studies” were selected or coded, and it does not appear that efforts at assessing inter-rater reliability of coding were made.
Results
- Page 4 – sections 3.1 and 3.2 –information on the development of the ACE-IQ does not seem to reflect the results of methods applied by the authors (although it is difficult to assess given the opacity of the methods section). This level of detail was missing from the Introduction – I recommend moving this section there.
- Pages 4-5 – Section 3.3 - This summary of articles is quite unsystematic and the studies reflected in the earlier systematic review are not described individually at all. Typically, a review focuses on summarizing and synthesizing existing evidence (e.g., by making a table of main findings, tabulating evidence in support of and/or against a given hypothesis, etc.).
- I disagree with the authors’ repeated assertion that believing the ACE framework is useful (e.g., for screening migrant populations in the U.S. or for screening children globally) requires believing they are perfect (e.g., “the best possible indicators of childhood trauma,” “key to understanding health trajectories,” “the optimum way to achieve this end”). I agree that there are limitations to the ACE framework and that focusing on ACEs can come at a policy cost. However, psychology and public health decisions are often made by satisficing rather than maximizing, and ACEs may be both meaningful and imperfect.
- Sections 3.4 to 3.8 – It is not at all clear to me how this authorial argument (although at times compelling) fits under Results.
- Section 4: These are not truly case studies in the typical sense of the term. I do not think it is appropriate to apply to individual-level phenomenon (i.e., an individual ACE score) to a hypothetical child based on the description of an entire community. The narratives were not well-grounded in the themes of the paper and did not provide relevant details about age when discussing teenagers and young adults. These sections featured lengthy quotations that were not properly cited with page numbers. The tables presenting ACE constructs were confusing and incomplete (why were not all possible ACE-IQ items included?).
Discussion
- The discussion does not adequately synthesize information from the review of the literature, instead focusing on the case studies. Language is strong and not scientific (e.g., “incredibly large blind spots,” “bizarre”).
Minor comments:
- Compound adjectives like “cost-effective” and “resource-scarce” should be hyphenated.
- I recommend minimizing the use of an em dash setting off a new clause (e.g., “ – with intervention necessary”), as often the new clause was not well integrated into the sentence as a whole.
- Western is inconsistently capitalized.
- The authors sometime use “we” and sometimes “I.”
- I recommend streamlining and minimizing subheadings, particularly under section 3.
Author Response
Please see attachment, many thanks.

Reviewer 2 Report
The manuscript entitled "The global relevance of the Adverse Childhood Experience Framework: knowledge gaps and implications for policy makers" tackled an important topic. However, essential revisions are needed to improve the logical flow. Please see my specific comments below.
- Apparently, this is a review and has a strong focus on the instrument of ACE 12 International Questionnaire (ACE-IQ). However, the title does not reflect the two important features and may confuse the readers.
- In the Abstract, why does the author want to number the presentations? Moreover, why does the author specifically mention “Research from the USA”? Additionally, the author mentioned “The tool reflects prevalent concerns in Western cultures, but global challenges such as forced economic migration, famine and persecution are neglected”, but the author has reviewed some publications in countries not in the West in the main text.
- Sometimes, the author used “we” and sometimes used “I” in the manuscript, this is confusing.
- I cannot understand why the author has a subsection of “Expanded definitions in the ACE-IQ” in Results section. Given that the author has clearly indicated that she wants to assess the ACE-IQ in the Introduction, the terms defined by the ACE-IQ should be described in the Introduction for readers to better understand this instrument at the very beginning.
- The author has especially reviewed on some studies in the section of “Case studies from the literature”. However, it is unclear how she decided which studies should be reviewed in details. Moreover, the three studies reviewed by the author in details were old (published between 1998 and 2006). I wonder how the information in the old publications can be applied to the current ear. Indeed, these societies have been developed with laws and regulations being installed and implemented. Therefore, the Adverse Childhood Experience may be changed in the past 15 to 20 years.
Author Response
Please see attachment, many thanks.
